# Behaviour Classification on Giraffes (*Giraffa camelopardalis*) Using Machine Learning Algorithms on Triaxial Acceleration Data of Two Commonly Used GPS Devices and Its Possible Application for Their Management and Conservation

**DOI:** 10.3390/s21062229

**Published:** 2021-03-23

**Authors:** Stefanie Brandes, Florian Sicks, Anne Berger

**Affiliations:** 1Institut für Biochemie und Biologie, University of Potsdam, Am Neuen Palais 10, 14469 Potsdam, Germany; st-brandes@web.de; 2Leibniz-Institute for Zoo- and Wildlife Research, Alfred-Kowalke-Straße 17, 10315 Berlin, Germany; 3Tierpark Berlin-Friedrichsfelde GmbH, Am Tierpark 125, 10319 Berlin, Germany; f.sicks@tierpark-berlin.de

**Keywords:** giraffe, triaxial acceleration, machine learning, random forests, behavior classification, giraffe conservation

## Abstract

Averting today’s loss of biodiversity and ecosystem services can be achieved through conservation efforts, especially of keystone species. Giraffes (*Giraffa camelopardalis*) play an important role in sustaining Africa’s ecosystems, but are ‘vulnerable’ according to the IUCN Red List since 2016. Monitoring an animal’s behavior in the wild helps to develop and assess their conservation management. One mechanism for remote tracking of wildlife behavior is to attach accelerometers to animals to record their body movement. We tested two different commercially available high-resolution accelerometers, e-obs and Africa Wildlife Tracking (AWT), attached to the top of the heads of three captive giraffes and analyzed the accuracy of automatic behavior classifications, focused on the Random Forests algorithm. For both accelerometers, behaviors of lower variety in head and neck movements could be better predicted (i.e., feeding above eye level, mean prediction accuracy e-obs/AWT: 97.6%/99.7%; drinking: 96.7%/97.0%) than those with a higher variety of body postures (such as standing: 90.7–91.0%/75.2–76.7%; rumination: 89.6–91.6%/53.5–86.5%). Nonetheless both devices come with limitations and especially the AWT needs technological adaptations before applying it on animals in the wild. Nevertheless, looking at the prediction results, both are promising accelerometers for behavioral classification of giraffes. Therefore, these devices when applied to free-ranging animals, in combination with GPS tracking, can contribute greatly to the conservation of giraffes.

## 1. Introduction

The complex importance of nature and biodiversity for human society is undisputed and biodiversity loss is considered a serious danger for the future of human beings [1,2]. Averting the dramatic loss of biodiversity and the subsequent loss of ecosystem services is possible through intensified conservation efforts, especially for keystone species [3]. Giraffes (*Giraffa camelopardalis*) are large, long-lived, non-territorial, browsing ruminants [4]. By covering long distances [5,6] while spreading seeds of different plants as well as by feeding on *Acacia* species [4], which play a role in the bush encroachment [7], they have an important part in sustaining Africa’s ecosystems [8]. Due to habitat loss and poaching, giraffe populations have declined by 30–40% over the last 30 years, with ongoing decrease in some subspecies. Therefore, according to the IUCN Red List (www.iucnredlist.org, accessed in December 2020) the category of the species *Giraffa camelopardalis* was changed from ‘least concern’ to ‘vulnerable’ in 2016 [9].

As an animal’s behavior depicts its response to the (changing) environment it is a “fundamental part of its biology” [10] and knowledge about animal behavior in the wild plays an important role in species conservation [11]. Understanding individual behavior enables us to illustrate its state of well-being [12], analyze the habitat use [13] and predict certain responses to varying influences [14], for example. Hence, monitoring the behavior of wildlife could improve suggestions for and predictions about their conservation management and help in assessing the success and failure of certain management measures [15].

The complexity of ecological processes and the behavior of wildlife under natural conditions can hardly be reconstructed in the laboratory [16]. Thus, methods to investigate wildlife behavior should be applied under field conditions which, in turn, face methodological challenges. With a mean female home-range size of approximately 90 km^2^ [17,18,19] giraffes, for example, are hard to follow and observe in their natural habitat. More recently, amassing research on remote monitoring of animal behavior and physiology using biologgers have removed many former limitations of observational studies [20]. Nowadays, additional sensors incorporated into GPS tags provide insight into the fine scale behavior of the study animals [21]. Unlimited by vegetation density, terrain, climate, observer bias or the scale of an animal’s space use, it is now possible to record the behavior of free-living wild animals by tagging them with high-resolution accelerometers.

Triaxial accelerometers attached to an animal, record motion in three-dimensional axes (dorsoventral, anterior–posterior and lateral) [10,22]. Many studies have shown that acceleration data can be used to infer the behavior of animals by employing various supervised machine learning algorithms [23,24]. To train the algorithms for pattern recognition and data classification, the acquisition of acceleration data is coupled with direct observation of the behaviors of the tagged animals. Using one portion of this observation-truthed data set to train the algorithm and another portion to infer behavior from it, allows validation of the predicted behavior [25]. This technique of behavioral detection by analyzing acceleration data has been successfully used on a variety of bird species [23,26,27,28,29,30,31], on marine animals [32,33,34,35,36,37], and on terrestrial mammals [25,38,39,40,41,42,43,44,45].

Due to their peculiar anatomy, physiology and sensitivity to drugs, giraffes are among the most challenging mammals to immobilize them, and thus to tag them with biologgers [46]. In the field, different attachment options have been used to follow giraffes with the help of GPS devices [5,6,47]. Accelerometers were tested for the first time in the Brookfield Zoo, USA, in 2015/2016 [48]. However, behavior was only classified as activity or inactivity [48]. The aim of our study was to attach and test GPS accelerometer devices to tame captive giraffes, to determine their efficacy for use on giraffes in the wild. Here, we analyzed triaxial acceleration data together with parallel observation data in easy machine learning algorithms to find species-specific behavioral classification models that could also be used in later studies on free-living, unobservable giraffes. We used two different commercially available accelerometers (one with a high and one with low resolution/measurement interval) in parallel on the study animals to find differences in sensor performances and suitability for behavioral classification. With this study we hope to establish a basis in the field of automatic behavior classification on giraffes in order to contribute to the species’ conservation.

## 2. Materials and Methods

### 2.1. Study Area and Animals

The study was conducted on three individuals of the species Giraffa camelopardalis.

In the African Lion Safari, a partly walk-, partly drive-through park in Ontario (https://lionsafari.com/, accessed in December 2020; 1386 Cooper Road, Hamilton, Ontario, Canada) the study was carried out on two female Rothschild’s giraffes (*Giraffa camelopardalis rothschildi*), namely Farrah (born on 28 January 2003) and Jaffa (born on 12 January 2005). Both were part of a herd with, in total, 12 individuals: two male adults, one young male and nine adult females, sharing their 24 ha outdoor enclosure with species such as white rhinos (*Ceratotherium simum*), Barbary sheep (*Ammotragus lervia*), or ostriches (*Struthio camelus*) and staying in a barn during the nights. Being a drive-through area, the outdoor enclosure included concrete driveways, but also grass and partly sandy underground. Food was supplied every day in form of hay in stationary, elevated feeding grounds, branches with leaves and sometimes carrots. Drinking water was possible *ad libitum* at a large pond in the northwest and a smaller pond in the south of the area or, after rain, from smaller puddles. The enclosure included flat, but also steep areas.

In the Zoological Garden Berlin (https://www.zoo-berlin.de/, accessed in December 2020; Hardenbergplatz 8, 10,787 Berlin, Germany) the study was carried out on one male Rothschild’s giraffe (*Giraffa camelopardalis rothschildi*). Max (24 July 2011) shared his 1770 m^2^ outdoor enclosure with a five-year-old male reticulated giraffe (*Giraffa camelopardalis reticulata*) and Defassa waterbucks (*Kobus ellipsiprymnus defassa*) and stayed in a barn during the nights. Hay and leaved branches were provided daily in stationary, partly elevated feeding stations. Drinking water was supplied *ad libitum* in an automatic water drinker attached 3 m above the ground and in a tray on a ground. The outdoor, flat enclosure consisted mainly of places with harder sand and a little grass, but partly also of concrete spots.

### 2.2. Accelerometers (e-obs and Africa Wildlife Tracking) and Collaring

For this study, two different triaxial accelerometers were used: e-obs (e-obs GmbH, Munich, Germany, www.e-obs.de, accessed in 2020) and AWT (Africa Wildlife Tracking, www.awt.co.za, accesses in 2020). Both sensors are a fixed part within proprietary GPS devices for wildlife, and measure their movement in three axes, sway (lateral), surge (anterior-posterior) and heave (dorso-ventral), which are perpendicular to each other. The axes’ fixation leads to a change in orientation when shifting the accelerometers’ position.

The e-obs device measured the data in user-set time intervals (bursts) [49]: we set a recording interval of 20 s and bursts with a frequency of 33.8 Hz for 2.43 s resulting in bursts of 82 data points per axis and recording interval, which were stored on the tag as unit-free digital numbers between 0 and 4095. The start and end of a burst were indicated by a pinger signal which was received through a communications receiver (YAESU VR-500). In the e-obs accelerometer, the *x*-axis measures the heave, the *y*-axis the sway and the *z*-axis the surge movement (Figure 1a). The data were stored on board and could be downloaded by radio with an e-obs BaseStation as a logger (.bin) file. To get the raw outputs, a data decoder (e-obs GmbH Wireless dataloggers decoding program, version 7.3) was used for converting and saving the data as a text (.txt) file. From there they could be further processed for the analyses.

The AWT device is a UHF-transceiver and measures one acceleration data point per axis continuously with a set frequency (here called “heartbeat”) of 1 Hz (minimum possible heartbeat) [50]. The AWT data could not be stored on the device but had to be directly transmitted to a laptop via an UHF-transceiver, an antenna (built by AWT) and the AWT app (version 2.0.6.129). With this, the data—unit-free digital numbers between −32,000 and +32,000—were stored in the app from where they could be downloaded and saved as an Excel file. In the AWT accelerometer, the *x*-axis measures the surge, the *y*-axis the sway and the *z*-axis the heave movement (Figure 1b). As the data of the AWT accelerometer were directly stored in an app, they did not need to be converted but could directly be imported to Excel.

For the female giraffes both accelerometers were tested at the same time by taping the e-obs device to the AWT harness, resulting in a total weight of 0.962 kg. On the male giraffe only the e-obs device was used, which was taped to a leather harness, leading to a final weight of 0.423 kg. The allowed total weight of such attachments to an animal is restricted to 2–5% of that animal’s body mass [51]. In all cases of this study, though, there was no possibility of measuring the giraffes’ exact body mass. Adult males and females can weigh about 970–1950 kg and 450–1200 kg, respectively [52]. Considering the minimum values, the used devices and harnesses weighed 0.04% and 0.21% of the male’s and females’ body mass, respectively.

Both harnesses were fixed on the giraffes’ heads between the ossicones and the medium lump (Figure 1). During behavior observation days, the devices were attached every morning before releasing the individuals into the outside areas and detached every evening after leading them back into their barns. In the male’s case the harness stayed on his head during the days and nights of the sleeping behavior observations. In this study, there was no need of sedation as all giraffes were trained several months in order to get used to the devices’ attachement and detachment.

### 2.3. Behavioral Observations

In order to allocate behaviors to the recorded data, we conducted behavior observations while the accelerometers were attached to the giraffes. The observations took place during the daytime over a period of14 days between 20 September and 11 October 2017 in Canada and over 11 days between 24 April and 14 May 2018 in Germany. In order to observe sleeping behavior, for three nights from 3–6 June 2018, we installed a Tayama 8.0 mm monochrome camera in the barn where the male giraffe Max spent its nights. During these three nights Max was also equipped with the e-obs accelerometer. For the e-obs device each behavior during a burst, the start and end time of which was signaled by a pinger, was written down in an observation protocol. As the AWT was set to measure the acceleration data every second, the protocol here included columns for the start and end time of a certain behavior. For a better overview another column was included to mark those behaviors occurring between two e-obs bursts. To synchronize the latter, the time of the AWT data and the observations, a radio-controlled alarm clock (TechLine TT-803) was used to get the exact times.

Since the accelerometers were attached to the giraffes’ heads, only behaviors whose movements were also transferred to the posture and movement of the head could be measured. Accordingly, protocol of the behavioral observation always included the height of the head as well as the corresponding behavior. The detailed, originally recorded behaviors (such as eating leaves at eye level, eating with the head stretched upwards, chewing changing to eating above eye level) were summarized into the following 14 behavioral categories, while only 12 could be included into the analyses and not all behaviors could be observed in sufficient sample size for each giraffe: standing (STA); lying (LIE); feeding above eye level (FEA); feeding at eye to middle level (FTM); feeding at deep level (FED); feeding at ground level (FEG); rumination (RUM); drinking (DRI); walking/slow- to medium-speed locomotion (WAL); running/high-speed locomotion (RUN); grooming (GRO); socio-positive behavior (SOC); socio-negative behavior/fighting (FIG); and sleeping (SLE). The single behavior category and head/neck level definitions are given in Table A1 and Figure A1, respectively.

### 2.4. Data Processing

For initial data processing the software Excel^®^ 2007 (version 12.0.4518.1014) was used. Recorded e-obs bursts and AWT acceleration data were listed and related to the corresponding observed behavior category. As the AWT sensor recorded data continuously, but as it was not possible to continuously fill out the protocol for observed behaviors, we separated the AWT data into behavior sections, similar to the e-obs data, but with different time spans for each single section (in the following also called “bursts”). For the further processing, only bursts with pure behavior categories—meaning only one category occurred from the start to the end of this burst—were used and all recordings with mixed behaviors within one burst were excluded from the analysis data set. For further analyses, out of the whole data pool, we randomly chose an equal number of data sets (*n* = 30; for Jaffa’s individual analysis *n* = 40) per behavior category and animal in order to avoid possible bias in the further analyses due to different occurrence amount of behavior categories.

With the switch of the e-obs tag from the giraffes in the Canadian zoo to the giraffe Max in Berlin’s zoo the position of the device varied due to its different attachment harness. Hence, the axes’ orientation and thus acceleration values slightly changed. To ensure comparability between the e-obs data of all three giraffes, Max’ values were adjusted: due to only small value variations within the behavior “standing without any movement” (within STA), the mean values per axis were calculated for that behavior for every giraffe. Afterwards, the differences per axis between the individual mean values were used for the adjustment of all recorded data. The corrected values were then used for all further analyses.

For further supervised machine learning analyses and for each axis separately we calculated 21 predictor variables out of the corrected, unit-free e-obs acceleration data of each burst using the Software R (version 3.4.3): arithmetic mean per axis (mnx, mny, mnz), standard deviation per axis (sdx, sdy, sdz), inverse coefficient of variation (ICVx, ICVy, ICVz), weighted mean of the frequency spectrum per burst and axis (wmx, wmy, wmz), kurtosis per axis (Kx, Ky, Kz), skewness per axis (Sx, Sy, Sz),and from a combination of all three axes: square-root-of-the-sum-of-squares (q) [23], pitch (p), and roll (r) [53]. In the same way, we calculated predictor variables out of the raw, unit-free AWT data. However, as here the behavior categories’ sequence lengths of those differed, the weighted mean (wm), influenced by the burst length, could not be included into the analysis, resulting in 18 predictors. The values for pitch (p) and roll (r) were given in the raw AWT outputs. As the way of computation was unknown, though, for comparability they were calculated in the same way as for the e-obs data.

### 2.5. Data Analysis

Data analysis was conducted with the software R (version 3.4.3). In a pilot study we tested the following supervised machine learning algorithms as well as a majority vote model: the linear discriminant analysis (LDA) [54], classification and regression trees (CART) [55], random forest (RF) [56], and support vector machines (SVM) [57] are common analyzing tools [23]; the *k*-nearest neighbor (KNN) approach [58] is an additional accurate way to analyze behavior data [24]. Out of all these machine learning algorithms, the RandomForest package [56] showed the highest overall accuracies; thus, only the results of this algorithm are presented here.

First, the (i) individual and (ii) combined data sets were analyzed. For the latter, we combined the corrected individual data of all giraffes to two single data sets—one for e-obs, one for AWT -and randomly chose 30 bursts per behavior category per data set from those. We analyzed both (i) and (ii) with a leave-one-out cross-validation: here, N–1 bursts (meaning: rows of predictor variables) were used to train the model, the left out (remaining) burst was tested against all others. This was performed for every vector of predictors, meaning that every row of the predictors was used *N*–1 times for the training and only once for the testing phase [59]. Then, in a (iii) cross-validation, we used the combined e-obs data of two giraffes and the AWT data set of one giraffe, respectively, for the training phase. The data set of the left-out giraffe was used for the testing phase. Here, only the results of (i) and (iii) will be taken into consideration: the individual analyses (i) show the general potential of applying supervised machine learning algorithms for the species-specific behavior classification on giraffes, whereas the cross-validations (iii) depict the potential of classifying an unknown data set of wild-ranging giraffes by training an algorithm with a labeled data set (the study’s data) [39,60].

For demonstrating the algorithm’s accuracy the predictability metrics ‚accuracy‘: proportion of all the predictions that are correctly classified, positively or negatively (1), ‚precision‘: proportion of predictions that are assigned to a behavior and really are the behavior (2), and ‚recall‘: proportion of possible predictions that can be assigned to a behavior and are correctly predicted as the behavior (3) [61] were calculated per tested animal and category:(1)$$\mathrm{accuracy}=\frac{\mathrm{truepositives}+\mathrm{truenegatives}}{\mathrm{truepositives}+\mathrm{truenegatives}+\mathrm{falsepositives}+\mathrm{falsenegatives}}$$
(2)$$\mathrm{precision}=\frac{\mathrm{truepositives}}{\mathrm{truepositives}+\mathrm{falsepositives}}$$
(3)$$\mathrm{recall}=\frac{\mathrm{truepositives}}{\mathrm{truepositives}+\mathrm{falsenegatives}}$$

The overall predictability metrics per individual were then defined as the mean of all the behavior categories’ results. In a final step, the Random Forest’s predictabilities of the e-obs and the AWT data were compared with the paired *t*-test (t.test(paired = T)).

## 3. Results

### 3.1. Application of the e-obs and AWT Accelerometer

#### 3.1.1. Battery and Storage Capacity

For the e-obs accelerometer recording time spans per day can be set. This is a great advantage if a study is only interested in activities at certain times of a day, thereby extending the lifespan of the battery. The device came with a battery lifetime and storage capacity calculator table. Here, values such as battery capacity in mAh, number of recording axes, burst length, frequency, amount of pinger signals etc. were typed in/calculated. With this study’s settings, the calculations resulted in a full (64 MB-) memory after 129 recording days and an empty battery after 42 recording days. However, though fully loaded before the study in Canada and turning the device off after every study day and back on the mornings of the day, the battery of the device still died after 14 recording days. In order to test and ensure the functionality on the study site, we conducted pilot recordings and data download events, which could have had an impact on the battery. Additionally, the device had already been used for several studies beforehand, which could also have affected the longevity of the battery. For the first period in Canada over 100,000 data rows were stored on the tag; however, due to the empty battery, it is hard to predict how many more data could have been stored on the memory. For the second study period in Germany, a different device was used; its battery lasted until the end of the study.

The AWT device was switched from an off status to a standby modus in August 2017. After this, we conducted a couple of pilot recordings in order to test the functionality of the device, to find the right settings and assess the axes’ orientation. In contrast to the e-obs accelerometer, the AWT, once started, cannot be switched off completely. Not customized yet for the continuous recording of accelerometer data, there were no calculations or suggestions from the producers for the battery’s lifetime. Though the device switched to a battery saving mode when not recording, the battery was empty after 17 full recording days (around 9–10 weeks after switching the device on). Unlike the e-obs accelerometer, the AWT’s battery cannot simply be reloaded as it is completely built into the collar, which can only be opened by the producers. This is why the accelerometer could not be used for the analysis on Max in Germany. A disadvantage of the device for the application in the wild is that the recorded data could not be saved on board but had to be directly transferred in real time to an app (see Section 2.2). However, no limits in the data capacity are known.

#### 3.1.2. Data Transmission

As described above, the recorded e-obs acceleration data were directly stored on board and could later be downloaded. With this, data recording is independent to weather, vegetation etc. Thus, during data recording there is no need for scientists or observers to be close by the animals wearing the accelerometers. Only during data download the tagged animals needs to be closer to the downloading e-obs BaseStation. Our study on kept animals always delivered the possibility of being close enough to the tag for the data download. In the wild, however, the data transmission distance between tag and BaseStation can be around 500 m to several kilometers, depending on the habitat.

The AWT data had to be transmitted through an antenna (UHF receiver) to an app on a laptop. Though it is said to be able to overcome a distance of a few hundred meters between UHF receiver and accelerometer [62], there were many interruptions in the data transmission during the study. Disruptions occurred for instance when the giraffe lowered its head and neck to the ground or when there were objects such as stone piles between the device and the antenna. Hence, the AWT shows not only disadvantages in recording data of hard to follow/observe species, but also comes with restrictions in the transmission when close to the animal. It is worth considering that during the study a pilot antenna, self-made by the AWT producers, was used. Applying a more developed transceiver might at least overcome the disruptions caused by objects.

### 3.2. Prediction Accuracy of Behavior Categories with Individual Analyses

The amount of recorded (“pure behavior”) data per behavior category and individual as well as those used for the analyses can be found in Table A1. The data from which the machine learning model was built is fully available on Movebank [63].

The prediction accuracies for the single individuals differed between the behavior categories as well as between the two different devices. They ranged from 61.7% for the category STA (standing) in the analysis for Farrah’s AWT data to even 100% for the category WAL (walking) in the analysis of Jaffa’s e-obs data (Table 1). In total, with the Random Forests algorithm the analyzed behaviors of the captive giraffes could be automatically classified to a high degree of accuracy (mean overall accuracy, precision, and recall for e-obs: 93.7%, 96.2%, and 96.1%, respectively; and for AWT: 90.4%, 93.7%, and 93.1%, respectively).

In general, STA (mean prediction accuracy for e-obs and AWT: 90.7% and 76.7%, respectively), SOC (e-obs and AWT: 83.5% and 89.7%, respectively) and RUM (91.6% and 86.5%) could be distinguished from the other categories at the lowest rates (Table 1). Within the e-obs data, they got often confused with FED (9 out of 100 times), FTM, RUM (each 5 out of 100) and LIE (4); FED (6 out of 30 times) and every other possible category except LIE (STA, FEA, FEG, RUM and WAL each 2 out of 30, FTM and DRI each 1 out of 30); and FTM (10 out of 100 times), STA (9), and LIE (5), respectively (Table A2, Table A3 and Table A4). Within the AWT data they got often confused with RUM (12 out of 70 times), FED (11), LIE (8) and FTM (6); FED (6 out of 30 times) and WAL (4); and STA (10 out of 70 times), FTM (7) and LIE (5), respectively (Table A5 and Table A6). Furthermore, LIE (mean prediction accuracy for e-obs and AWT: 95.3% and 85.6%, respectively), FTM (91.2% and 89.3%) and FED (91.3% and 88.4%) also had lower prediction accuracies than the categories not mentioned yet (Table 1). Within the e-obs analyses they got often confused with RUM (5 out of 60 times) and STA (3); RUM (9 out of 100 times), FEA (7), FED (5), WAL and SOC (3 each); and FEG (14 out of 100 times), FTM and WAL (3 each), respectively (Table A2, Table A3 and Table A4), while the AWT analyses showed confusions with RUM (6 out of 30 times) and STA (4); WAL (6 out of 70 times), RUM (5), FED (4) and SOC (3); and FEG, SOC (each 5 out of 70 times) and WAL (3), respectively (Table A5 and Table A6).

All the categories that could be predicted less accurately included a wide range of different movements as well as head and neck positions on different levels. SOC, for instance, could only be evaluated in sufficient amounts for Farrah, including a wide variety of behaviors, such as rubbing or just sniffing on another individual. STA combines fine-scaled behaviors like standing still without further movement of any body part (e.g., for vigilance behavior) as well as standing while moving the head (scanning). At the same time, there are behavior categories with similar movement patterns: when ruminating, a giraffe’s movements can resemble chewing food while holding the neck on different levels, which was included in the FTM category, and vice versa. When feeding from the ground, the patterns can look similar to those of drinking and vice versa. Hence, the value boarders of the single axes per category highly overlap with those of other categories, making possible confusions between certain behavior categories more likely.

On the contrary, FEA (mean prediction accuracy for e-obs and AWT: 97.6% and 99.7%, respectively), SLE (97.1% for e-obs), DRI (96.7% and 97.0%) as well as WAL (97.6% and 95.0%) could each be well distinguished from other categories (Table 1). For the e-obs data, these categories were mainly confused with FTM (3 out of 60 times) and SOC (1); STA (4 out of 30 times); FEG (4 out of 30 times); and FED (4 out of 100 times) and FTM (3), respectively (Table A2, Table A3 and Table A4). For the AWT data (Table A5 and Table A6), FEA has not been wrongly predicted as another category once whereas DRI only got confused with FEG (4 out of 30 times) and FED (1) and WAL with FED (4 out of 70 times) and FTM (4).

Compared to other behaviors, these better predicted categories included less fine-scaled behaviors and more unique value patterns: e.g., by feeding on leaves that were above the giraffe’s own height, as the individual has to lift its head. For feeding from the ground (mean prediction accuracy for e-obs and AWT: 95.6% and 96.0%, respectively; Table 1) or drinking a giraffe lowers its head and neck to the ground. Compared to a giraffe’s head and neck level while standing still, for example, this leads to a shift of the accelerometer’s values to lower (e-obs)/higher (AWT) levels (FEA) or vice versa (FEG/DRI).

Amongst others, possible restrictions of accelerometers could be depicted in the analysis of the SLE category: with a prediction accuracy of 97.1% (e-obs;Ta) it could be well distinguished from other categories. However, as described above, it could not be predicted 100%, but got wrongly identified as STA 4 out of 30 times (Table A2). In this study, SLE includes the REM sleep—which should be classified easily due to its unique body posture (lying down, neck bending to the back)—and sleeping with the neck and head up, which could be the cause for the misclassifications by having a similar body/neck and head posture to STA.

### 3.3. Prediction Accuracy of Behavior Categories with Cross-Validations

The prediction accuracies for the cross-validations, like for the individual analyses, differed between the behavior categories as well as the two different devices. They ranged from only 50.0% for the category RUM (rumination) to 97.6% for the category FEG (feeding from the ground), both in the analysis for Farrah’s AWT data as test data set (Table 2). In total, with the Random Forests algorithm the analyzed behaviors of the captive giraffes could be automatically classified to a high degree (mean overall accuracy, precision, and recall for e-obs: 88.4%, 92.8%, and 91.7%, respectively; and for AWT: 81.5%, 83.1%, and 79.0%, respectively). However, compared to the individual analyses (Table 1), the prediction accuracies as well as precision and recall for the cross-validations were lower. When training the algorithm with the data of two giraffes, it is possible that the third, tested giraffe—which behaviors should be predicted—includes characteristics that did not occur in the other individuals.

Though the results differed between the e-obs and AWT device, categories with a high variation in fine-scaled behaviors and movement patterns such as STA (mean prediction accuracy for e-obs and AWT: 91.0% and 75.2%, respectively), FTM (83.0% and 76.0%, respectively), FED (84.7% and 84.0%) and RUM (89.6% and 53.5%) could generally be predicted at lower rates than FEG or WAL, similar to the individual analyses (3.2.). With the e-obs data they got mainly confused with FED (7 out of 90 times), FTM and RUM (each 6); RUM (14 out of 90 times), WAL (10) and FED (8); FEG (15 out of 90 times) and WAL (10); and FTM (15 out of 90 times) and RUM (9), respectively (Table A7, Table A8 and Table A9). With the AWT data they got mainly mixed up with FED (10 out of 60 times), FTM (9) and RUM (6); FED (9 out of 60 times), STA (7) and RUM (6); FEG (9 out of 60 times) and STA (4); and FTM (28 out of 60 times) and STA (27), respectively (Table A10 and Table A11).

At the same time, though sometimes being a (wrong) prediction outcome for other categories (see above), FEG (mean prediction accuracy for e-obs and AWT: 91.3% and 95.8%, respectively) could generally be predicted the most accurate for the cross-validations and got confused with mainly FED (13 out of 90 times for e-obs; 3 out of 60 times for AWT), while WAL (90.6% and 91.4%) got wrongly predicted mainly as FTM (11 out of 90 times for e-obs; 6 out of 60 times for AWT) and FED (3 out of 90 and 10 out of 60, respectively) (Table A7, Table A8, Table A9, Table A10 and Table A11).

### 3.4. Comparison of Prediction Accuracies of Behavior Categories with the e-obs and AWT Accelerometer

In addition to the prediction analyses, the accuracies of both accelerometers were compared for the analyses of the Canadian giraffes. Both devices could deliver data leading to high predictabilities, precisions and recalls (Table 1 and Table 2) which were comparable to, if not better than in other studies.

For both Farrah and Jaffa, there were no significant differences in the results for the e-obs and AWT data. Still, there was a tendency that the categories could be predicted better with the e-obs than with the AWT data sets (paired *t*-test, Farrah: *t* = 1.50, *p* = 0.097; Jaffa: *t* = 1.98, *p* = 0.052). For some behavior categories, the AWT data could lead to higher prediction accuracies than the e-obs accelerometer. This is the case for those categories that could generally be better predicted than other categories and which lead to a great shift in the head movement, hence, the accelerometer values. Those include FEA (mean predictions accuracy for the individual analyses: 99.7% for AWT vs. 97.6% for e-obs), DRI (97% vs. 96.7%) or FEG (96.0% vs. 95.6% and 95.8% vs. 91.3% for the cross-validations) (Table 1 and Table 2). The overall accuracies, however, tended to receive higher percentages with the e-obs than with the AWT data (mean overall accuracy, precision, and recall for e-obs vs. AWT for the individual analyses (Table 1): 93.7% vs. 90.4%, 96.2% vs. 93.7% and 96.1% vs. 93.1%, respectively; and for the cross-validation analyses (Table 2): 88.4% vs. 81.5%, 92.8% vs. 83.1% and 91.7% vs. 79.0%, respectively).

## 4. Discussion

Biological sensor tags, such as triaxial accelerometers, are often used to study animal behavior and could contribute immensely to the field of conservation [21].In this study, two different commercially available high-resolution accelerometers (e-obs and AWT) were tested on three captive giraffes in order to analyze the accuracy of an automatic behavior classification for this species, here showing and discussing the results for the machine learning algorithm Random Forests.

### 4.1. Random Forests Machine Learning Algorithm for Automatic Behavior Classification

As mentioned, when analyzing the acceleration data of the observed giraffes with several machine learning models beforehand, the random forests model (RF) could outperform the other algorithms tested. This matches findings of Nathan and colleagues (2012), who show that with an accuracy of 91%, the RF performed best for identifying behaviors in griffon vultures, compared to the LDA, CART, SVM and ANN (artificial neural network) [23]. In our study, the RF not only delivered good results for the accuracies but also showed high overall precision and recall values. Precision and recall concentrate more on the ratio of true positives (a category was correctly identified as this category) and showed in this study that the RF to a high proportion correctly classified the categories and made less mistakes in assigning a category wrongly to the “goal category”.

The advantage of the RF is its use of multiple decision trees, unlike the CART, which only creates one final decision tree—setting the final class due to a majority vote. With this, the RF reduces the influence of a single tree’s possible misclassification [64]. Hence, when examining giraffe data of the two accelerometers used here, we recommend applying the RF for ensuring the most accurate analysis of the data.

### 4.2. Performance Depending on Input Data (Individual vs. Cross-Validation Analyses)

The performance of the RF differed depending on the input data: for both e-obs and AWT accelerometer, the individual analyses could deliver higher results compared to the cross-validations. With the leave-one-out cross-validation, an individual’s data are used as training and test data at the same time. Therefore, the whole variety within the behavior categories of a single giraffe can be captured, which makes it easier to assign new data points to the classes (categories) of the training data. However, different animals, even though of the same species, can show different movement patterns. For humans [65,66] or elephants [39], it was found that using data from multiple people or animals decreased the predictability of behaviors compared to using only data of the same individual.

Nevertheless, particularly with the possibility of employing these devices on wild giraffes the results of the cross-validations should be of high interest because the cross-validation method reflects the situation of training an algorithm with a labeled data set (the data of our) and by this classifying the unknown data of wild-ranging giraffes [39,60]. Furthermore, with an overall accuracy of up to 91.5% (e-obs, Jaffa’s test data; Table 2) the cross-validation analyses could still deliver good results. However, the study’s small sample size (three and two tested giraffes for e-obs and AWT, respectively) needs to be taken into account; conducting a triaxial acceleration data study on wild giraffes should also comprise the analysis of several free-living individuals. 

Investigating wild animals, natural behaviors that usually are not or are rarely observed in zoo-kept conspecifics such as mating or hunting behavior must be considered. These behaviors as well as measurements of mixed behavior bursts might occur in the wild, but they will not be classified because the algorithms were not trained on them. These undetected behaviors could be narrowed down through a probability threshold and classified as “unknown behavior” [25]. Anyway, testing the unlabeled data of numerous wild giraffes against the patterns of our study’s few individuals surely will lead to a higher percentage of behavior misclassifications.

Covering more than three individuals for our study could lead to lower prediction accuracies than shown here. However, an ideal model would be trained on more data; thus, with more observational data or more individuals, we would be able to train a more robust model. This is why we recommend a follow-up study that can combine our study’s data with that of new individuals, in order to analyze the impact of including more giraffes on the behavior categories’ predictability. However, many studies in the field of wildlife ecology and conservation research face the problem of small sample sizes. Species may either be hard to observe or assess (e.g., due to remote or impassable habitats or an elusive life) or simply because of being too rarely distributed or barely zoo-kept. However, species-specific inter-behavior variance in the body movements for behaviors such as resting, feeding, or locomotion usually is higher than intra-behavior variance. This makes it possible that, in principle, models based on few individuals can also provide reliable behavioral predictions. Rast et al. [25] were able to infer the behavior of wild foxes using acceleration data, a supervised machine learning algorithm (ANN), and a training data set obtained from only two captive individuals.

### 4.3. Prediction Accuracy of and Confusions between Behavior Categories

For both accelerometers, all behavior categories could be predicted well, considering the highest accuracy per behavior (Table 1 and Table 2). Nevertheless, behaviors that involve more fine-scale movements and a higher variety of body postures or head and neck levels (such as STA, FTM, and SOC) were harder to distinguish than those with less activity and/or a smaller variety of the neck’s and/or head’s position (such as FEA, FEG, and DRI).The fact that similar movements or postures in different behavior categories can have a negative influence on the differentiation between them has already been considered as problematic for several other species [39,43,67]. 

An ability to determine sleep in a giraffe could be key for this species’ conservation [68]. During the phase of REM sleep a giraffe is fully unprotected [4] and the sleep duration can depict its stress exposure and be a response to a changing food quality/availability or to other stressors [69]. However, as Martiskainen and colleagues could find for cows [67] and Gerencsér et al. for dogs [70], their body posture and that of the accelerometer did not greatly differ while standing or lying. For both standing still as well as sleeping, the e-obs device’s axes of the giraffe Max nearly showed no amplitude due to the lack of acceleration. Therefore, the height difference might not have been detected as there were no differences in the gravitational forces, which are the only values being measured by the device during such steady positions [67]. This confusion of behaviors (STA vs. SLE) could be a restriction in correctly identifying giraffe sleeping places, though results for SLE still were high with an overall accuracy of 97% in Max’ individual analysis.

For deploying this study’s devices on wild giraffes some important things have to be taken into account. During a recording burst, a variety of at least two different behavior types can occur, for example if the giraffe is changing its behavior from ruminating to walking, from drinking to standing to walking etc. With that, it could be harder to assign the unknown data point of the wild giraffe to a known category of this study’s individuals: as their data only included bursts with a single behavior this could lead to more misclassifications and decreasing prediction accuracies. Choosing an appropriate frequency of sampling and short intervals between recording bursts of short length could avoid many recordings of mixed behaviors [22], and the same settings for train and test data set had to be used. Another possibility for facing this problem could be the promising variable-time segment/window analyses. In this approach segment lengths are chosen due to significant changing points in the data [71]. However, this method requires continuous data. The AWT was supposed to deliver that precondition by measuring the device’s position every second, but could not, amongst others due to transmission interruptions for one to several seconds or even minutes from time to time.

### 4.4. Comparison of e-obs and AWT Analyses and Handling

Giraffes in the wild, of course, cannot be trained for attachment of devices like the captive animals of this study. As the devices were not delivered with an automatic detachment mechanism, sedation would need to be organized for both the attachment and detachment of the devices on a wild giraffe.

For both e-obs and AWT accelerometer the giraffe behaviors could be better predicted in the individual analyses than in the cross-validations. Though there was a tendency of the e-obs delivering higher prediction accuracies, precisions as well as recalls, there was not a significant difference between the two accelerometers’ results. Hence, both delivered promising results.

Nonetheless, both accelerometers come with slightly different restrictions that have to be taken into account before deploying them in the wild. Depending on the study’s task and conditions, the device battery life can be a challenge: for an AWT device deployed on elephants, recording a total of 920 GPS locations on an eight-hourly schedule, acceleration data were not recorded, the battery lasted 428 days [72]. However, for the AWT device future studies about the battery lifetime with different settings of recording acceleration data are needed as additionally recording of acceleration data will lead to a slightly faster drain of the battery. E-obs devices were usually used on birds or relatively small wild mammals [73,74,75] but are adaptable to the particular study in terms of battery used: here, solar-buffered versions or larger batteries (with a longer life span) are possible options.

However, for all devices, the following is valid: battery size, mass, and thus capacity are limited by the size of the tagged animal; the total weight of the device must not exceed 5% of the animal’s body mass [51]. Wireless connection to satellites (necessary for measuring GPS positions) and to other devices (such as BaseStations for data download) are particularly energy-intensive, so the number of GPS location measurements and data download events are a major determinant of battery life. In contrast, accelerometry is not based on radio connection to other technical devices and, therefore, measurement and storage even of the high-resolution triaxial acceleration data only consumes a minor amount of energy, but it occupies a great amount of storage capacity due to the large amount of data. Nevertheless, in accelerometry the higher the sampling frequency, the higher the energy and storage consumption and the more frequent data download is needed consuming also extra energy. Decreasing the sampling frequency, however, can decrease the behaviors’ predictability as was shown for the Great sculpin (*Myoxocephalus polyacanthocephalus*) [76]. For diminishing this issue, e-obs devices record high-resolution acceleration data using bursts which decreases the received data volume and storage consumption compared to continuously recording [22]. 

A major disadvantage of the AWT device was its storage capability: while GPS data can be stored on the tag and downloaded at a later point, the acceleration data for this study could only be recorded and stored on the tag every 10 min, together with the GPS coordinates, or had to be transmitted directly to a laptop in order to save the values for the chosen frequency (here minimum heartbeat of 1 s). This, of course, is not useful for the wilderness as the transceiver has to be continuously connected to a tracker app via the laptop and within a few 100 m distance to the accelerometer [62], most probably affecting the animals’ behavior [77]. Additionally, during the study it was recognized that the data transmission could be interrupted due to larger piles of stone or small hills being between the tag and the transceiver or while the giraffe bent down its neck and head to drink or feed from the ground.

### 4.5. Future Accelerometers for Studies on Giraffes

As the axes of rotation are fixed within the device the orientation must be taken into account before deploying a device. If attaching devices differently to different individuals, the values always need to be adjusted accordingly before comparing or using this study’s results with or for newly collected accelerometer data. One option to avoid this requirement is to use always the same kind of collars and harnesses and attach them to giraffes. However, the future trend will be towards small, solar-powered devices that can be fixed to the ossicone as shown in a recent study by Hart and colleagues [78]. The e-obs GmbH already developed small solar-powered tags for birds [49] which also could be used on giraffes, but the attachment method still is a highly and critically discussed topic [78]. Future research still needs to concentrate on attaching such devices in a fixed position for all individuals, without the possibility of accidental shifts and without harming an individual. If this is given for the attachment of e-obs devices, only our study’s data would need an adjustment to other devices’ axes-orientation once and could then be used for all future analyses of different individuals.

Another great progress would be the use of solar-powered devices such as the e-obs bird loggers [49]. Automatically reloading during day, solar-powered devices are less dependent on the battery drain. Furthermore, limited storage capacities could be overcome by regularly downloading the data with an e-obs BaseStation by regular aerial tracking flights [79] or by building permanently installed download stations on fixed spots, i.e., waterholes, which will be regularly visited by giraffes.

## 5. Conclusions

In summary, both the e-obs and the AWT device can be promising accelerometers for the behavior classification on giraffes (*Giraffa camelopardalis*). Nevertheless, the e-obs device could provide better conditions than the AWT, not only in terms of data storage and battery life—which can even be improved by the methods mentioned above—but also in terms of size. A disadvantage still remains regarding the burst method. Certain behaviors, which occur between two bursts, could be missed and only continuous data could be used for the already mentioned variable-time segment/window analyses [71]. Nevertheless, getting a general overview of the behavioral patterns at certain locations could also be achieved with the burst measurements. Which settings (frequency, burst length etc.) to use should depend on the study topic and species, though [22].Furthermore, we recommend conducting a future study to analyze the actual impact on the prediction accuracies when testing a data set with mixed behaviors against a labeled train data set with pure behaviors during one burst, when using the same methods as in this study.

In conclusion, where direct behavior observations of wild giraffes face restrictions caused by darkness or rough terrain, as in a recent study about the sleep of Angolan giraffes (*Giraffa camelopardalis angolensis)* [68], the use of accelerometers is a helpful tool to observe giraffe behavior continuously. Combining different methods—such as recording the GPS location, acceleration data, conducting observations where possible—and different studies’ results can gather the needed information about the giraffe biology in order to protect and conserve the species.

## Figures and Tables

**Figure 1 sensors-21-02229-f001:**
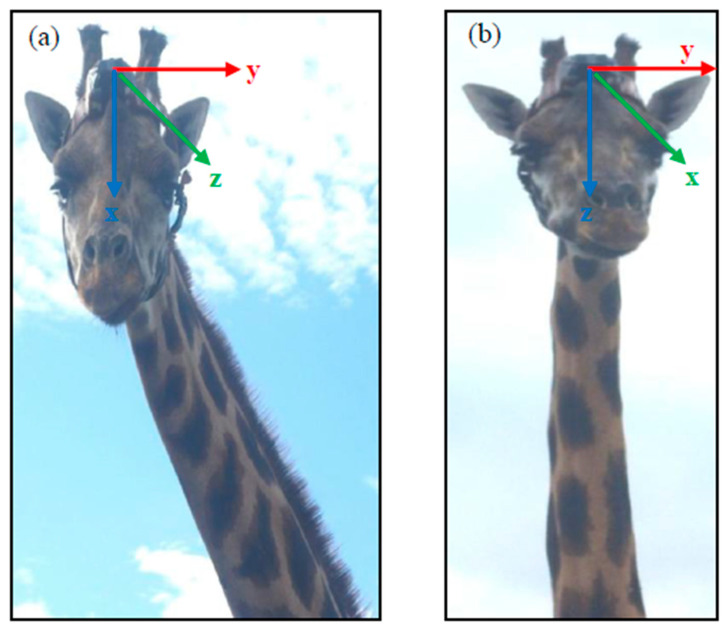
Attachment of the accelerometers, shown on the female giraffes, with the axes’ orientation for the (**a**) e-obs and (**b**) AWT accelerometer; red: sway (lateral movement), green: surge (anterior-posterior movement), blue: heave (dorso-ventral movement).

**Table 1 sensors-21-02229-t001:** Accuracy per behavior with overall accuracy (mean), precision, and recall ^1^ for the RF algorithm analyses of the individual e-obs and AWT data of all three giraffes; STA: standing, LIE: lying, FEA: feeding above eye level, FTM: feeding at eye to middle level, FED: feeding at deep level, FEG: feeding at ground level, RUM: rumination, DRI: drinking, WAL: walking, SOC: socio-positive behavior, FIG: socio-negative behavior/fight, SLE: sleep. Behaviors with a burst amount *n* < 30 per giraffe have not been analyzed (‘---’). Highest and lowest (i) mean accuracy per behavior category per device, (ii) overall accuracy (mean), (iii) precision, and (iv) recall are marked in green and red, respectively.

Behavior Category/Accuracy	e-obs			Mean Accuracy per Behavior Eobs	AWT		Mean Accuracy per Behavior AWT
	Max	Farrah	Jaffa		Farrah	Jaffa	
STA	0.879	0.877	0.965	0.907	0.617	0.916	0.767
LIE	0.993	0.913	---	0.953	0.856	---	0.856
FEA	0.969	0.983	---	0.976	0.997	---	0.997
FTM	0.922	0.879	0.936	0.912	0.865	0.921	0.893
FED	0.930	0.904	0.904	0.913	0.861	0.906	0.884
FEG	0.983	0.951	0.933	0.956	0.969	0.951	0.960
RUM	0.957	0.833	0.959	0.916	0.779	0.951	0.865
DRI	---	0.967	---	0.967	0.970	---	0.970
WAL	0.967	0.962	1	0.976	0.944	0.955	0.950
SOC	---	0.835	---	0.835	0.897	---	0.897
FIG	0.957	---	---	0.957	---	---	---
SLE	0.971	---	---	0.971	---	---	---
mean	0.953	0.910	0.949	0.937	0.875	0.933	0.904
precision	0.973	0.946	0.968	0.962	0.915	0.959	0.937
recall	0.972	0.942	0.968	0.961	0.905	0.957	0.931

^1^ overall accuracy (mean): average of the proportion of all the predictions that are correctly classified, positively or negatively; precision: average of the proportion of predictions that are assigned to a behavior and really are the behavior; recall: average of the proportion of possible predictions that can be assigned to a behavior and are correctly predicted as the behavior [61].

**Table 2 sensors-21-02229-t002:** Accuracy per behavior with overall accuracy (mean), precision, and recall ^1^ for the RF algorithm analyses of the cross-validated e-obs and AWT data of all three giraffes; e.g., Max test (e-obs): Max’ individual e-obs data set tested against the combined e-obs train data sets of Farrah and Jaffa, e.g., Farrah test (AWT): Farrah’s individual AWT data set tested against the AWT train data set of Jaffa; STA: standing, FTM: feeding at eye to middle level, FED: feeding at deep level, FEG: feeding at ground level, RUM: rumination, WAL: walking. Highest and lowest (i) mean accuracy per behavior category per device, (ii) overall accuracy (mean), (iii) precision, and (iv) recall are marked in green and red, respectively.

Behavior Category/Accuracy	e-obs			Mean Accuracy per Behavior e-obs	AWT		Mean Accuracy per Behavior AWT
	Max Test	Farrah Test	Jaffa Test		Farrah Test	Jaffa Test	
STA	0.867	0.899	0.963	0.910	0.727	0.777	0.752
FTM	0.838	0.781	0.870	0.830	0.723	0.796	0.760
FED	0.836	0.881	0.825	0.847	0.895	0.785	0.840
FEG	0.862	0.939	0.937	0.913	0.976	0.940	0.958
RUM	0.906	0.853	0.928	0.896	0.500	0.569	0.535
WAL	0.864	0.888	0.967	0.906	0.875	0.952	0.914
mean	0.862	0.874	0.915	0.884	0.827	0.803	0.815
precision	0.919	0.918	0.948	0.928	0.783	0.878	0.831
recall	0.898	0.913	0.939	0.917	0.779	0.801	0.790

^1^ overall accuracy (mean): average of the proportion of all the predictions that are correctly classified, positively or negatively; precision: average of the proportion of predictions that are assigned to a behavior and really are the behavior; recall: average of the proportion of possible predictions that can be assigned to a behavior and are correctly predicted as the behavior [61].

## Data Availability

The data from which the machine learning model was built is fully available on Movebank (www.movebank.org, accessed on 1 December 2021) under the study name “Accelerometry Giraffes”.

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
