# Peer review of "Behaviour Classification on Giraffes (Giraffa camelopardalis) Using Machine Learning Algorithms on Triaxial Acceleration Data of Two Commonly Used GPS Devices and Its Possible Application for Their Management and Conservation"

_sensors, 2021, doi:10.3390/s21062229_

Round 1

Reviewer 1 Report

This paper describes results of two different accelerometer tags fixed to captive giraffes and how they compare to direct behavioral observations and to each other. The authors discuss the pros and cons of each tag, and the limitations of their use on wild giraffes. The manuscript could use quite a bit of editing, both for English language and in general to make meaning more clear and more concise. The authors should address more fully how their extremely small sample size might have affected their results. The paper is interesting and important in that it is testing and validating new technology that could be of great use to understanding giraffe behavior and ecology. However, due to the short battery life and transmission difficulties neither device seems appropriate for use on wild giraffes, and this should be made more explicit. The authors should add recommendations of what is needed for future development of a device that may be more suitable.

The authors describe that the tags were fitted on to harnesses and that training was needed for the giraffe to accept this. Wild giraffes would not receive this training, and it is not clear from this paper how long a harness would last on a wild animal. The authors also stated that they needed to correct data for device position between the male and the females – would this also need to be done for each individual? The authors should add more detail about how this could be assessed. Bursts, and how these were matched to behavior are not clearly detailed, and similarly how data was recovered from the eobs tag is not clear. Throughout it would be better to give the full behavior name, rather than the 3 letter acronyms. The detail of the predictions was difficult to read and should be made easier to follow, perhaps by a matrix table or something similar.

The manuscript needs editing for English language. There are many word choices or use of apostrophe that are jarring for a native speaker. In general the writing could be improved to be more clear. I have made some detailed comments where it is particularly unclear below.

L15 and L37 – change ‘nowadays’ to ‘current’ or ‘today’s’.

L21 – Names of eobs and AWT should be given in full.

L36-39 – 2 sentence paragraphs are unnecessary.

L93 and elsewhere – remove the asterisk in front of the date.

L95 and elsewhere – but outer do you mean outdoor?

L101 – by plane do you mean flat?

L131 – what do you mean by exemplary? Do you mean using females as an example? What is meant by sway, surge, and heave? Figure headings should stand alone without need for reference to the text.

L135 – change ‘to’ to ‘two’.

L137 – what do you mean by fixedly built? Rephrase.

L228-232 – it is not clear whether you tested all these algorithms.

L324-339 – this hard to read paragraph might be better presented as a matrix table.

L347 – what is meant by recall?

Reviewer 2 Report

This paper will be useful for wildlife researchers.  It will provide scientists with useful information about technology for obtaining behavior information.

Line 21 - There needs to be a clear statement in the abstract that devices were attached to the top of the head.

Line 173 - It is really important to train the animals to tolerate the device.  This needs more emphasis.  Habituated animals will behave differently than non-habituated animals.

Line 208 - The problem of the device moving after attachment changes the data.  In your study, you knew it had moved and you made adjustments.  This reviewer is concerned that data would not be accurate if the device moved and the researchers were not aware of it.  If multiple devices are used, it should be emphasized that the same harnesses design should be used to attach them to the heads of different animals.  This will prevent changes in the position of the device on the animal's head from confounding the data obtained from it.

Line 515 - Change "lasting time" to battery life.

Line 522 - Provide an estimate of a long time for battery life.  If this reviewer was using this device, I want an accurate estimate of how long the battery would last.  Provide some estimates of battery life at different settings for the two devices.  This is information that would affect a researcher's decision on how the device would be used.  In many research applications, short battery life would limit its use.
